# Transient rapamycin treatment can increase lifespan and healthspan in middle-aged mice

Alessandro Bitto[1†], Takashi K Ito[1†], Victor V Pineda[1], Nicolas J LeTexier[1], Heather Z Huang[1], Elissa Sutlief[1], Herman Tung[1], Nicholas Vizzini[1], Belle Chen[1], Kaleb Smith[1], Daniel Meza[1], Masanao Yajima[2], Richard P Beyer[3], Kathleen F Kerr[4], Daniel J Davis[5], Catherine H Gillespie[5], Jessica M Snyder[6], Piper M Treuting[6], Matt Kaeberlein[1*]

[1]Department of Pathology, University of Washington, Seattle, United States; [2]Vaccine and Infectious Disease Division, Fred Hutchinson Cancer Research Center, Seattle, United States; [3]Department of Environmental and Occupational Health Sciences, University of Washington, Seattle, United States; [4]Department of Biostatistics, University of Washington, Seattle, United States; [5]Department of Veterinary Pathobiology, University of Missouri, Columbia, United States; [6]Department of Comparative Medicine, University of Washington, Seattle, United States

**Abstract** The FDA approved drug rapamycin increases lifespan in rodents and delays age-related dysfunction in rodents and humans. Nevertheless, important questions remain regarding the optimal dose, duration, and mechanisms of action in the context of healthy aging. Here we show that 3 months of rapamycin treatment is sufficient to increase life expectancy by up to 60% and improve measures of healthspan in middle-aged mice. This transient treatment is also associated with a remodeling of the microbiome, including dramatically increased prevalence of segmented filamentous bacteria in the small intestine. We also define a dose in female mice that does not extend lifespan, but is associated with a striking shift in cancer prevalence toward aggressive hematopoietic cancers and away from non-hematopoietic malignancies. These data suggest that a short-term rapamycin treatment late in life has persistent effects that can robustly delay aging, influence cancer prevalence, and modulate the microbiome.

*For correspondence: kaeber@ uw.edu

†These authors contributed equally to this work

Competing interests: The authors declare that no competing interests exist.

## Introduction

Successful interventions that increase healthy longevity in people could have profound benefits for quality of life, productivity, and reduced healthcare costs (*Goldman et al., 2013*; *Kaeberlein et al., 2015*). The drug rapamycin is a promising candidate for such an intervention, as it has been shown to increase lifespan in numerous species (*Johnson et al., 2013*) and to delay or reverse multiple age-associated phenotypes in mice including cognitive decline (*Halloran et al., 2012*; *Majumder et al., 2012*), cardiac dysfunction (*Dai et al., 2014*; *Flynn et al., 2013*), immune senescence (*Chen et al., 2009*), and cancer (*Anisimov et al., 2011*). Recently, a six week treatment with the rapamycin derivative RAD001 was reported to improve immune function in elderly people, as measured by response to influenza vaccine (*Mannick et al., 2014*), suggesting that at least some of the effects on aging in mice are conserved in humans. Despite these impressive results, the utility of rapamycin or other mTOR inhibitors to delay aging may be limited by side effects. The high doses of

**eLife digest** Old age is the single greatest risk factor for many diseases including heart disease, arthritis, cancer and dementia. By delaying the biological aging process, it may be possible to reduce the impact of age-related diseases, which could have great benefits for society and the quality of life of individuals. A drug called rapamycin, which is currently used to prevent organ rejection in transplant recipients, is a leading candidate for targeting aging. Rapamycin increases lifespan in several types of animals and delays the onset of many age-related conditions in mice.

Nearly all of the aging-related studies in mice have used the same dose of rapamycin given throughout the lives of the animals. Lifelong treatment with rapamycin wouldn't be practical in humans and is likely to result in undesirable side effects. For example, the high doses of rapamycin used in transplant patients cause side effects including poor wound healing, elevated blood cholesterol levels, and mouth ulcers. Before rapamycin can be used to promote healthy aging in humans, researchers must better understand at what point in life the drug is most effective, and what dose to use to provide the biggest benefit while limiting the side effects.

Now, Bitto et al. show that treating mice with rapamycin for a short period during middle age increases the life expectancy of the mice by up to 60%. In the experiments, mice were given two different doses of rapamycin for only three months starting at 20 months old (equivalent to about 60-65 years old in humans). After receiving the lower dose, both male and female mice lived about 50% longer than untreated mice, and showed improvements in their muscle strength and motor coordination. When given the higher dose, male mice showed an even greater increase in life expectancy, but the female mice did not. These female mice had an increased risk of developing rare and aggressive forms of blood cancer, but were protected from other types of cancer.

Both drug treatments also caused substantial changes in the gut bacteria of the male and female mice, which could be related to effects of rapamycin on metabolism, immunity and health. More studies are needed to uncover precisely how such short-term treatments can yield long-term changes in the body, and how such changes are related to lifespan and healthy aging.

rapamycin and its derivatives used clinically to prevent organ transplant rejection are associated with adverse events, including impaired wound healing, edema, elevated circulating triglycerides, impaired glucose homeostasis, gastrointestinal discomfort, and mouth ulcers (*Augustine et al., 2007*; *de Oliveira et al., 2011*). While many of these side effects have not been observed in mice at the lower doses that extend lifespan, chronic treatment with encapsulated rapamycin (eRapa) in the diet at 14 ppm has been reported to cause gonadal degeneration in males, increased risk of cataracts, and impaired response to a glucose tolerance test (*Wilkinson et al., 2012*; *Lamming et al., 2012*).

## Results

Based on the premise that transient treatment with rapamycin during middle-age might be more suitable for clinical efforts to promote healthy aging than continuous treatment throughout life, we set out to investigate whether a single three-month treatment regimen can extend lifespan and healthspan in C57BL/6JNia mice starting at 20–21 months of age. We initially used a treatment regimen consisting of intraperitoneal (i.p.) injections of 8 mg/kg rapamycin daily for 90 days. This dose was selected because we have previously found that it increases survival and alleviates disease phenotypes in short-lived mouse models of dilated cardiomyopathy, muscular dystrophy, and the severe mitochondrial disease Leigh Syndrome (*Ramos et al., 2012*; *Johnson et al., 2013*). Based on efficacy in the Leigh Syndrome mouse model and serum drug levels in wild type mice, we estimate that this treatment regimen is comparable to dietary delivery of eRapa at approximately 378 ppm (*Johnson et al., 2015*), or 27-fold higher levels than initially shown to extend lifespan in mice when continuous treatment is initiated at either 9 months or 20 months of age (*Harrison et al., 2009*; *Miller et al., 2011*). Because prior studies have noted differences in the magnitude of lifespan extension following continuous rapamycin treatment in male versus female animals (*Harrison et al., 2009*; *Miller et al., 2011*, *2014*), we examined the effect of this regimen in both sexes independently.

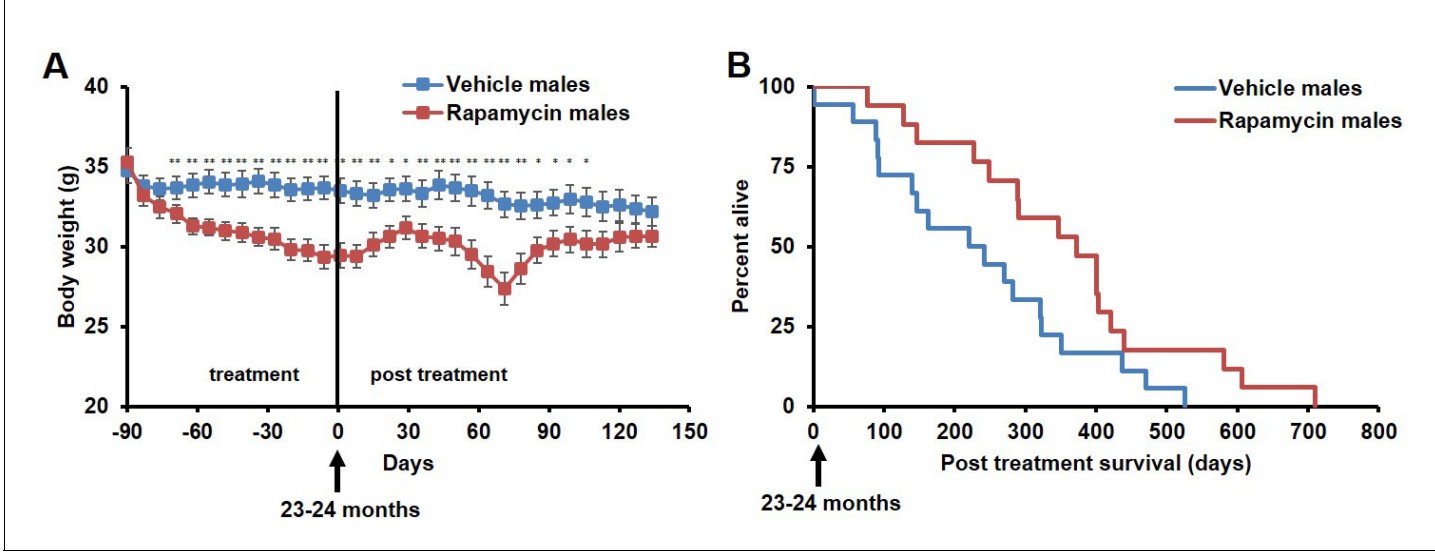

**Figure 1.** Rapamycin injection at 8 mg/kg/day for 3 months extends life expectancy of male mice. (**A**) Body weight of male mice measured weekly after starting rapamycin and vehicle treatment. Data are indicated as mean ± s.e.m. *p<0.05. **p<0.01. (**B**) Survival of control and rapamycin-treated male mice following the end of treatment. p=0.02. N=18 vehicle injected, N=17 rapamycin.

The following figure supplements are available for figure 1:

**Figure supplement 1.** Rapamycin serum level does not differ between female and male mice.

**Figure supplement 2.** Food intake of male mice receiving 8 mg/kg/day i.p. rapamycin or vehicle injections.

**Figure supplement 3.** Survival plots of male mice treated with 8 mg/kg/day i.p. rapamycin for 90 days starting around 600 days of age.

**Figure supplement 4.** Inclusion of non-age-related deaths does not alter survival outcomes.

Serum rapamycin levels did not differ significantly between male and female animals in our study (*Figure 1—figure supplement 1*).

During the three-month treatment period, we noted a significant decline in body weight of male mice receiving rapamycin injections relative to vehicle treated controls (*Figure 1A*), although food

**Table 1.** Sex-segregated comparison of median and mean post-treatment life expectancy for mice receiving rapamycin by injection (8 mg/kg/day) or feeding (128 ppm). M: males, F: females.

| | Median life expectancy (days) | Percent life expectancy increase | Mean life expectancy (days) | Percent life expectancy increase |
|---|---|---|---|---|
| Vehicle M | 231 | | 235 | |
| Rapamycin (8mg/kg/day) M | 372 | 61 | 358 | 53 |
| Vehicle F | 161 | | 177 | |
| Rapamycin (8mg/kg/day) F | 161 | 0 | 171 | -3 |
| Eudragit M | 193 | | 199 | |
| Rapamycin (126 ppm) M | 279 | 44.6 | 242 | 21 |
| Eudragit F | 184 | | 175 | |
| Rapamycin (126 ppm) F | 256 | 39 | 240 | 37 |

**Table 2.** Sex-segregated comparison of median and mean lifespan for mice receiving rapamycin by injection (8 mg/kg/day) or feeding (128 ppm). M: males, F: females.

|  | Median lifespan (days) | Percent median lifespan increase | Mean lifespan (days) | Percent mean lifespan increase |
|---|---|---|---|---|
| Vehicle M | 925 |  | 929 |  |
| Rapamycin (8mg/kg/day) M | 1054 | 14 | 1050 | 13 |
| Vehicle F | 847 |  | 858 |  |
| Rapamycin (8mg/kg/day) F | 847 | 0 | 853 | -1 |
| Eudragit M | 914 |  | 912 |  |
| Rapamycin (126 ppm) M | 1037 | 14 | 984 | 8 |
| Eudragit F | 879 |  | 883 |  |
| Rapamycin (126 ppm) F | 960 | 9 | 951 | 8 |

intake remained similar during the treatment (*Figure 1—figure supplement 2*). Decreased body weight persisted for several weeks following cessation of treatment (*Figure 1A*). This was accompanied by a striking increase in median life expectancy from the end of treatment of 60% (p=0.02, *Figure 1B*, *Table 1*) and an increase in overall median lifespan from birth of 16% (p=0.03, *Figure 1—figure supplement 3*, *Table 2*). This effect is larger than both the absolute and relative magnitude of lifespan extension resulting from continuous treatment to death with 14 ppm eRapa starting at around the same age in UMHET3 mice (*Harrison et al., 2009*). The longest-lived rapamycin-treated male in our cohort survived for 710 days post treatment to approximately 1400 days of age. Based on a survey of the literature, this is likely one of the longest-lived wild type C57BL/6 animals ever reported.

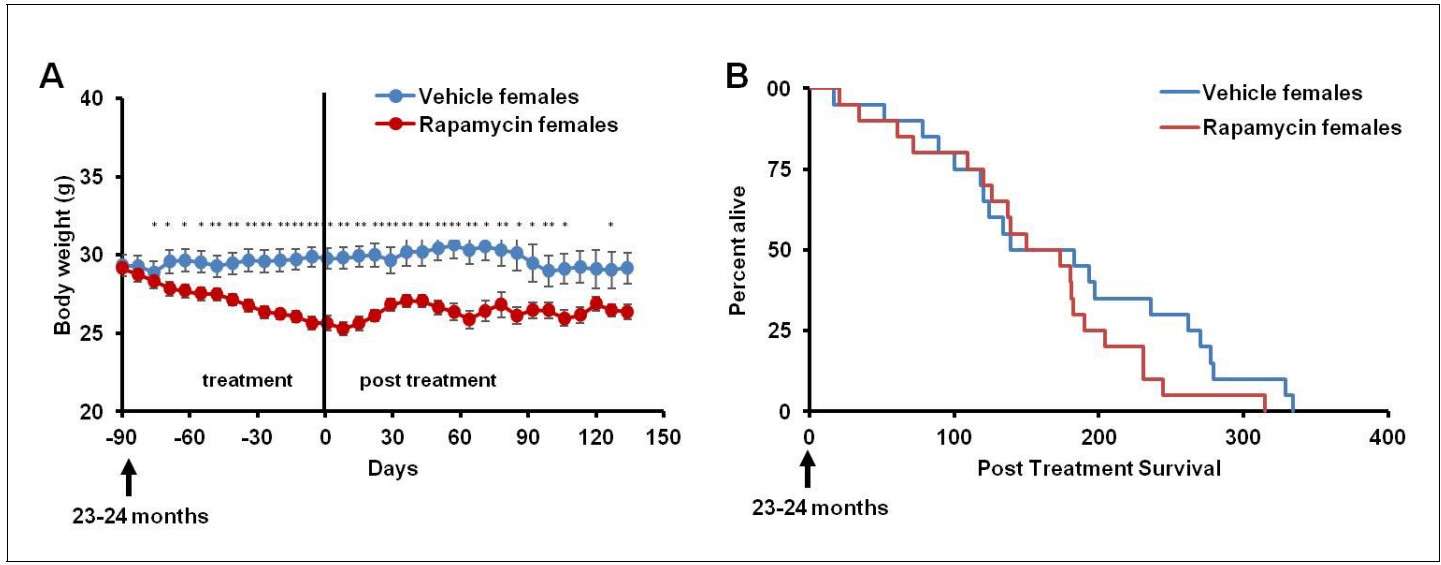

**Figure 2.** Rapamycin injection at 8 mg/kg/day for 3 months does not increase life expectancy of female mice. (**A**) Body weight of female mice measured weekly after starting rapamycin and vehicle treatment. Data are indicated as mean ± s.e.m. *p<0.05, **p<0.01 (**B**) Survival of control and rapamycin-treated female mice following the end of treatment. p=0.261. N=20 vehicle injected, N=20 rapamycin.

The following figure supplements are available for figure 2:

**Figure supplement 1.** Survival plots of female mice treated with 8 mg/kg/day i.p. rapamycin for 90 days starting around 600 days of age.

**Figure supplement 2.** Food intake of female mice receiving 8 mg/kg/day i.p. rapamycin or vehicle injections.

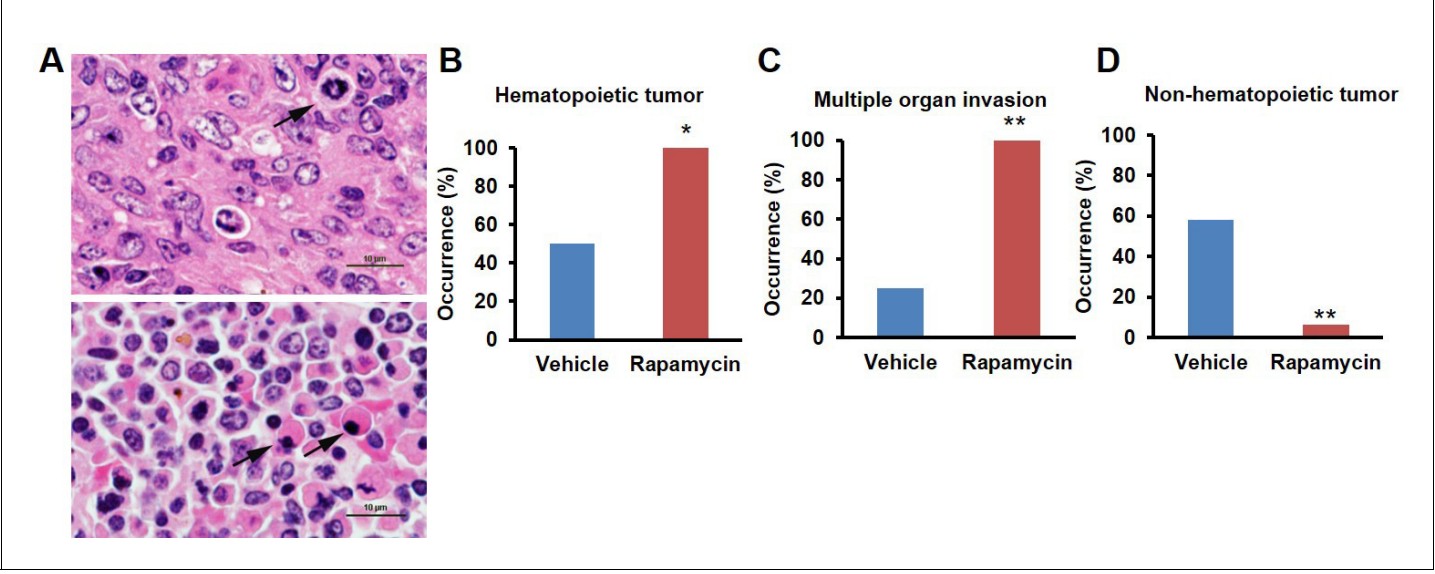

**Figure 3.** Rapamycin injection at 8 mg/kg/day for 3 months alters cancer incidence of female mice. (A) Hematoxylin and eosin (H&E) sections of multisystemic aggressive lymphoma (top) and atypical plasmacytoid lymphoma (bottom) from rapamycin-treated female mice. Arrows indicate a bizarre mitotic figure (top) and round cells with strongly eosinophilic cytoplasm (plasmacytoid morphology, bottom). Original magnification 60x. Bar = 10 μm. (B) Hematopoietic cancer incidence of rapamycin-treated (16 female) and vehicle-treated (12 female) mice. (C) Incidence of multiple organ invasion of hematopoietic tumors in rapamycin-treated (16 female) and vehicle-treated (6 female) hematopoietic tumor-bearing mice. (D), Non-hematopoietic cancer incidence of rapamycin-treated (16 female) and vehicle-treated (12 female) mice. *p<0.05. **p<0.01.

The following figure supplements are available for figure 3:

**Figure supplement 1.** Morphologies of aggressive hematopoietic tumors observed in rapamycin injected females.

**Figure supplement 2.** Cancer incidence of male mice with rapamycin injection at 8 mg/kg/day for 3 months.

This rapamycin regimen also resulted in a decline in body weight of female mice together with an increased trend of food intake in vehicle-treated animals with time (*Figure 2A*, *Figure 2—figure supplement 2*); however, we failed to detect a similar increase in post-treatment survival and lifespan in females (p=0.261, *Figure 2B*, *Figure 2—figure supplement 1*, *Tables 1–2*). Cox proportional hazards regression with robust standard errors and adjusting for cohort indicated significant evidence of a treatment difference between male and female mice (p=0.023). We speculated that this lack of lifespan extension results from an increase in aggressive hematopoietic cancers in the rapamycin treated females. Histopathological analysis showed that while 6/12 control females had round cell tumors (lymphoma and histiocytic sarcoma of hematopoietic origins), comparable to previous studies in C57BL/6 mice (*Blackwell et al., 1995*; *Treuting et al., 2008*), all the rapamycin-treated females examined (16 out of 16) had round cell tumors (*Figure 3A,B* and *Figure 3—figure supplement 1*, p=0.002). Additionally, an uncommon variant of lymphoma with plasmacytoid morphology affected 4 out of 16 mice examined in the rapamycin group and no vehicle treated mice (*Figure 3A* and *Figure 3—figure supplement 1A,B*). Round cell tumors affected more organs in rapamycin treated females, with multiple (≥2) organs affected in all of the rapamycin treated females compared to only 3 vehicle treated females (*Figure 3C*, p=0.01). Together, these data indicate a more aggressive phenotype of hematopoietic tumors in the high dose rapamycin treated females. In contrast, the incidence of non-hematopoietic neoplasms was dramatically decreased in the rapamycin group (*Figure 3D*). Nine non-hematopoietic neoplasms (5 pituitary adenomas, and 2 pulmonary adenomas, and 2 thyroid adenomas) were detected in 7 out of the 12 examined vehicle treated mice; whereas, only 1 out of 16 rapamycin treated females had non-hematopoietic neoplasia (1 pituitary adenoma) (*Figure 3D*, p=0.004).

**Table 3.** Sex-pooled comparison of median and mean post-treatment life expectancy for mice receiving rapamycin by injection (8 mg/kg/day) or feeding (128 ppm).

|  | Median life expectancy (days) | Percent life expectancy increase | Mean Life expectancy (days) | Percent life expectancy increase |
|---|---|---|---|---|
| Vehicle | 188 |  | 204 |  |
| Rapamycin (8mg/kg/day) | 231 | 22 | 257 | 26 |
| Eudragit | 190 |  | 188 |  |
| Rapamycin (126 ppm) | 270 | 42 | 257 | 37 |

Both control and rapamycin treated male cohorts had a high incidence of systemic round cell neoplasia which affected ≥2 organs, and had an intravascular component and a morphology most consistent with histiocytic sarcoma, although immunohistochemistry was not performed (*Figure 3—figure supplement 2*). Several male mice also had eosinophilic (hyaline) protein droplets in the renal tubules consistent with lysozyme accumulation associated with histiocytic sarcoma (*Hard and Snowden, 1991*). Of the examined vehicle treated males, all (14 out of 14) had hematopoietic neoplasia affecting at least one organ. Additionally, 7 other types of neoplasia were detected on histopathology from the vehicle treated male group, including 1 gastric neoplasia, 1 thyroid adenoma, and 5 pulmonary adenomas. One male mouse had a subcutaneous facial mass characterized by sheets of round cells, most consistent with an extramedullary plasma cell tumor or mast cell tumor. Rapamycin treated males also had a high incidence of systemic round cell neoplasia affecting at least one organ (9 out of 13). In the rapamycin treated male group, 12 additional non-round cell tumors were detected on histopathology, including 1 hepatocellular carcinoma, 2 intestinal adenomas, 1 splenic hemangiosarcoma, 5 pulmonary adenomas, 1 pulmonary carcinoma, 1 thyroid adenoma, and 1 Zymbal's gland adenoma (*Figure 3—figure supplement 2*). In the male group, no cases of systemic lymphoma with plasmacytoid morphology were seen. Although the number of mice examined is relatively small, based on these observations we speculate that this regimen of rapamycin treatment induced the detrimental side effect of aggressive hematopoietic cancers specifically in female mice, which occurred earlier in life and were sufficient to prevent lifespan extension in these animals despite other beneficial effects, such as reduced non-hematopoietic cancers. When data from both sexes are pooled together, daily injection of 8 mg/kg rapamycin for three months resulted in a non-significant (p=0.16) increase in life expectancy of 23% (*Table 3*). Between randomization and the end of treatment, 5 male mice (3 vehicle, 2 rapamycin) died of non-age related causes and were excluded from the analysis. Their inclusion in the analysis of survival does not significantly affect the results as described above (*Figure 1—figure supplement 4*).

In order to assess whether transient treatment with a lower dose and different delivery of rapamycin might reduce side effects in female mice and increase lifespan in both sexes, we utilized dietary eRapa at 126 ppm. Mice were fed a diet containing eRapa or the encapsulation control eudragit diet for 90 days starting at 20–21 months of age and then returned to a standard chow diet. In contrast with the effects of daily injection of 8 mg/kg rapamycin (*Figures 1A* and *2A*), body weight and food intake were largely unaffected by this dietary rapamycin regimen (*Figure 4 A,C,E*, *Figure 4—figure supplement 1*), except for a transient, small increase in body weight in female mice fed eRapa, as determined by Student's t test (*Figure 4E*). Median post-treatment life expectancy was significantly increased by 42% when female and male animals were considered together (p=0.002, *Figure 4B*, *Table 3*) and overall lifespan was increased by 13% (p=0.003, *Figure 4—figure supplement 2A*, *Table 4*). Similar results were observed stratifying on sex; three months of 126 ppm eRapa administration significantly increased post-treatment survival in both males and females independently (*Figure 4D,F*, *Figure 4—figure supplement 2B,C* and *Tables 1–2*). Cox proportional hazards regression with robust standard error estimates did not find evidence that sex modifies the treatment effect (p=0.904). Between randomization and the end of treatment, 1 male mouse died of non-age related causes and was excluded from the analysis. Its inclusion in the analysis does not significantly affect the results as described above (*Figure 4—figure supplement 3A,B*).

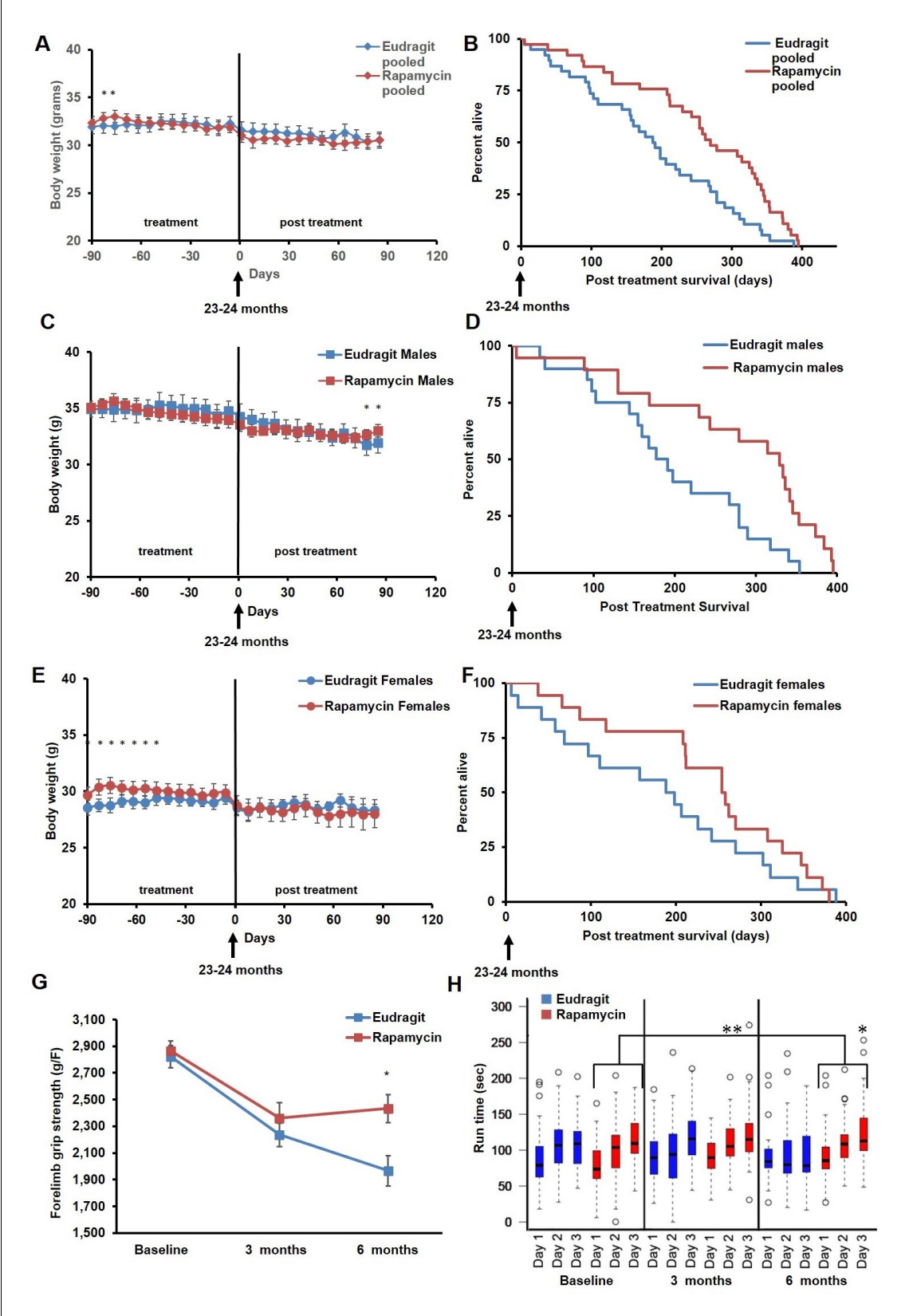

**Figure 4.** Rapamycin feeding at 126 ppm for 3 months extends life expectancy. (**A, C, E**) Body weight of (**A**) sex-pooled, (**C**) male, and (**E**) female mice measured weekly after starting rapamycin and eudragit treatment. *p<0.05. Survival of (**B**) sex-pooled control and rapamycin-treated mice following the
*Figure 4 continued on next page*

*Figure 4 continued*

end of treatment. p=0.002. N=38 vehicle injected, N=37 rapamycin. (C). Survival of (D) male, and (F) female control and rapamycin-treated mice following the end of treatment. (D) N=20 eudragit males, N=19 126 ppm eRapa males. (F) N=18 eudragit females, N=18 eRapa females. (G) Forelimb grip strength tests measured prior to treatment initiation (baseline), upon cessation of treatment (3 months), and 3 months after the drug withdrawal (6 months). (H) Rotarod performance tested prior to treatment initiation (baseline), upon cessation of treatment (3 months), and 3 months after the drug withdrawal (6 months). Data are plotted with box-whisker plots, showing median, 25th and 75th percentile, as well as individual outliers. Statistical significance was calculated with a linear mixed-effect model, using treatment group, measurement date, and measurement day as fixed effects and individual mice identifiers as random variables. *p<0.05 rapamycin vs. control at 6 months. **p<0.01 rapamycin at 6 months vs. rapamycin at baseline.

The following figure supplements are available for figure 4:

**Figure supplement 1.** Food intake of mice receiving 126 ppm eRapa or eudragit control.

**Figure supplement 2.** Effects of 126 ppm eRapa treatment on lifespan in male and female mice.

**Figure supplement 3.** Inclusion of non-age-related deaths does not alter survival outcomes.

---

In parallel with survival, we also examined healthspan parameters in these cohorts. The effects of rapamycin on age-associated decline in muscle function and motor coordination were assessed by testing forelimb grip strength and Rotarod performance. Assessments were performed in the same animals before onset of treatment, at the end of the 90 day treatment, and 90 days after the end of the treatment. While eudragit-fed mice showed a steady decline in both grip strength and Rotarod performance, rapamycin-fed mice scored significantly better than control mice in both assays after treatment (*Figure 4G,H*), suggesting that the healthspan promoting effect of rapamycin continues after treatment is discontinued. Furthermore, only rapamycin-fed mice showed a progressive increase in Rotarod performance during the 3 days of training and over the whole study, with mice performing better at 6 months compared to baseline (*Figure 4H*, large bracket).

In the course of routine animal husbandry, we noted that mice treated with either rapamycin regimen produced consistently smaller fecal pellets than age-matched controls (*Figure 5A–D*). This difference was present in both dry (*Figure 5C,D*) and freshly excreted feces (*Figure 5—figure supplement 1A*), indicating that water content is not a major factor affecting feces size. In addition, feces size was affected independently of the method of rapamycin delivery (*Figure 5C,D*), and the effect was persistent after cessation of treatment (*Figure 5—figure supplement 1B,C*). We hypothesized that changes in the microbiome may underlie this phenotype and therefore analyzed the fecal microbiome for each of the cohorts used in this study by deep-sequencing of bacterial 16S rRNA. Distance based permutation multivariate analysis of variance (MANOVA) (*Anderson, 2001*) indicated that rapamycin treatment induced a significant change in the composition of fecal microbiome (p=0.018 for injection cohorts. p=0.015 for feeding cohorts.p=0.005 for pooled cohorts), even after accounting for delivery method and batch effects (*Figure 6—figure supplements 1–3*).

Among the most notable changes in fecal bacterial DNA content seen in the global microbiome analysis was a significant increase in prevalence of segmented filamentous bacteria (SFB, *Candidatus*

---

**Table 4.** Sex-pooled comparison of median and mean lifespan for mice receiving rapamycin by injection (8 mg/kg/day) or feeding (128 ppm).

|  | Median lifespan (days) | Percent median lifespan increase | Mean lifespan (days) | Percent mean lifespan increase |
|---|---|---|---|---|
| Vehicle | 874 |  | 892 |  |
| Rapamycin (8mg/ kg/day) | 922 | 5 | 944 | 5 |
| Eudragit | 885 |  | 899 |  |
| Rapamycin (126 ppm) | 996 | 13 | 968 | 8 |

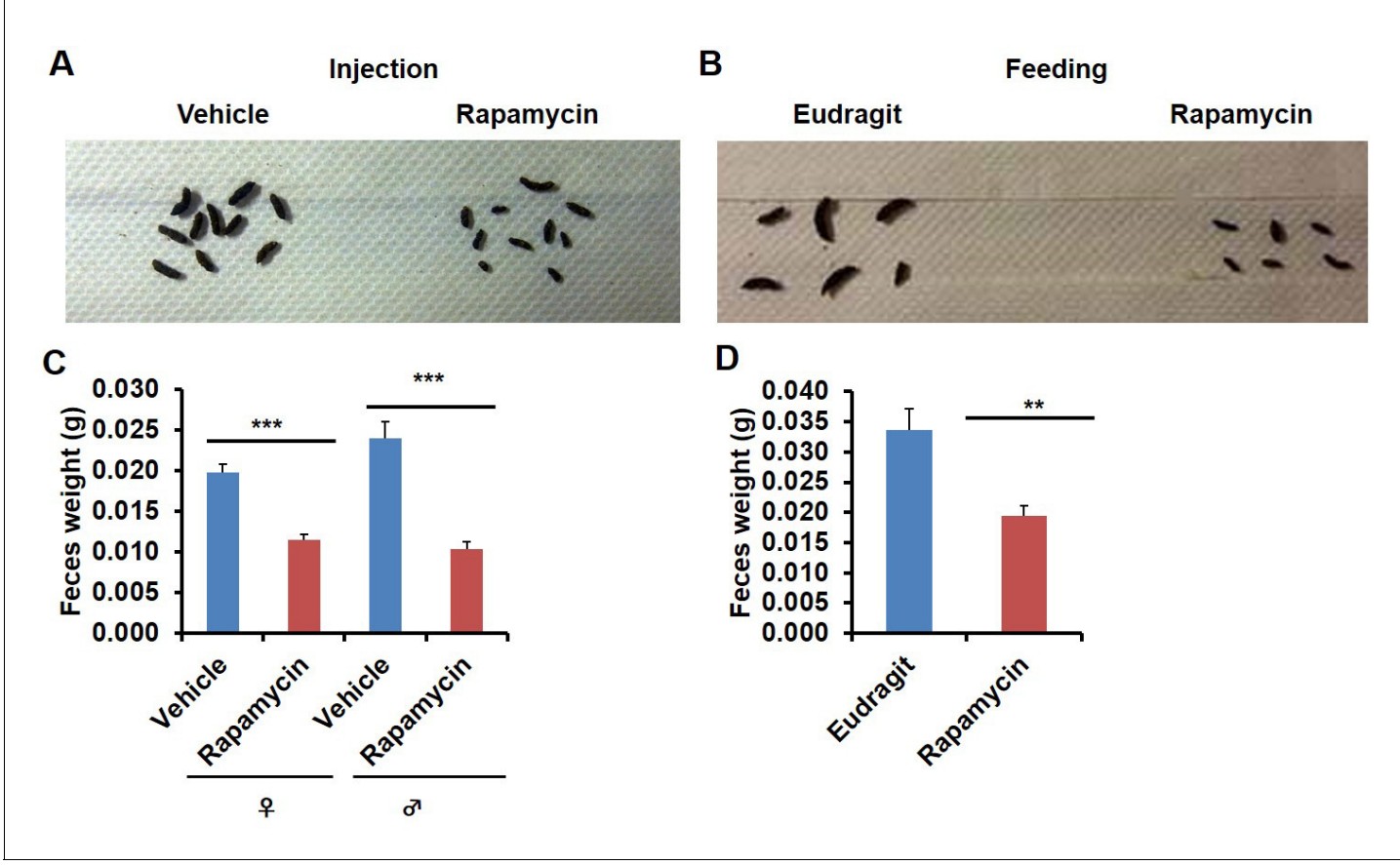

**Figure 5.** Rapamycin decreases fecal pellet size. (**A**) Photograph of feces collected from rapamycin-injected and vehicle-injected animals at 3 months of the treatment. (**B**) Photograph of feces collected from rapamycin-fed and eudragit-fed animals at 3 months of the treatment. (**C**) The weight of fecal pellets collected from rapamycin-injected and vehicle-injected animals at 3 months of the treatment. N = 22–24. (**D**) The weight of fecal pellets collected from rapamycin-fed and eudragit-fed female animals at 3 months of the treatment. N = 11. Data are indicated as mean ± s.e.m. *p<0.05. **p<0.01. ***p<0.001.

The following figure supplement is available for figure 5:

**Figure supplement 1.** Rapamycin decreases fecal pellet size persistently after cessation of treatment.

*Arthromitus* sp.) in the rapamycin treated animals (*Figure 6A*, *Figure 6—figure supplement 2A*). SFB are intestinal Gram-positive bacteria with a segmented and filamentous morphology, and are not normally present at high levels in aged mice (*Ericsson et al., 2014*). The SFB genome lacks a majority of virulence factors and SFB are not invasive (*Prakash et al., 2011*); however, their tight adhesion to the intestinal epithelial cell induces differentiation of host immune cells (*Atarashi et al., 2015*). The increase in SFB following rapamycin treatment was confirmed by real-time PCR of DNA from fecal samples obtained from both mice receiving injections or encapsulated rapamycin (*Figure 6B*, *Figure 6—figure supplement 2B*), as well as by semi-quantitative histological scoring of the small intestine in an independent cohort of mice obtained from the Harrison Lab at the Jackson Laboratory and injected with 8 mg/kg/day of for 3 months at the University of Washington (*Figure 6C,D*, *Figure 6—figure supplement 2C*). Since increased SFB DNA was observed both in mice injected with rapamycin and mice fed dietary eRapa, this effect is independent of mode of drug delivery. To the best of our knowledge, this represents the first pharmacological intervention to increase SFB in any animal. It will be of interest to determine whether these and other effects of rapamycin on the microbiome are shared across species and play any causal role in the beneficial or detrimental effects of this drug.

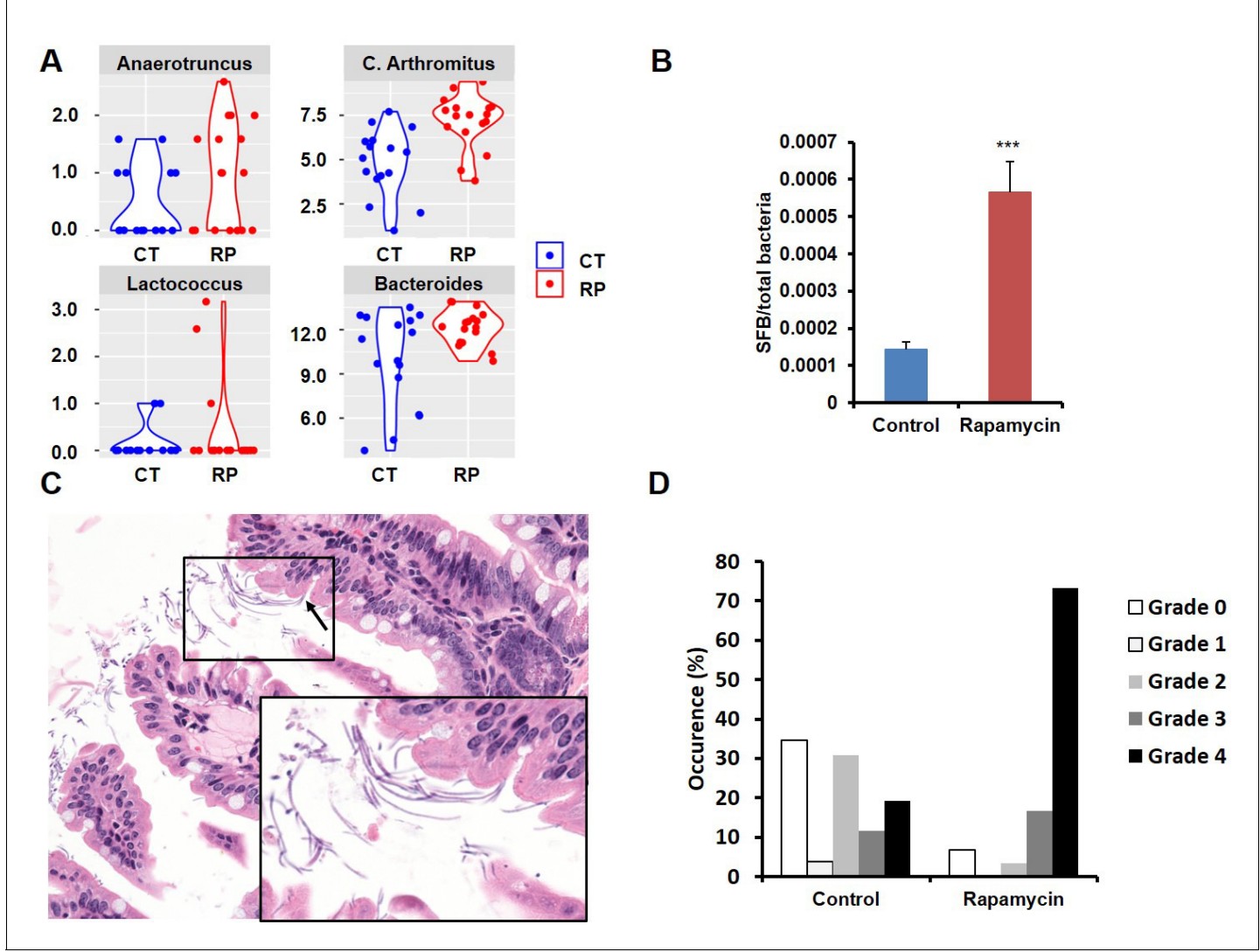

**Figure 6.** Rapamycin changes the composition of gut microbiota and increases segmented filamentous bacteria. (A) Violin plots representing the Log$_2$ 16S rRNA gene abundance for operation taxonomy units (OTUs) in fecal samples that are significantly differentially prevalent at a false discovery rate of 0.05 using the R package metagenomeSeq, controlling for the delivery method, gender and mouse batch effects. (C) Arthromitus (*Candidatus Arthromitus* sp.) refers to segmented filamentous bacteria (SFB). CT and RP indicate control and rapamycin, respectively. (B) The ratio of SFB DNA to total bacterial DNA in fecal samples measured by real-time PCR. N = 32. Data are indicated as mean ± s.e.m. ***p<0.001. (C) Representative H&E section of the intestine of rapamycin-treated mouse. Arrow indicates SFB attached to the intestinal epithelial cells. Lower right quadrant: magnification of area enclosed in black rectangle. (D) Semi-quantitative grading of SFB amount in the intestinal tissue section. 0 indicates the absence of SFB. 1–4 indicates the grades of the SFB amount with 1 lowest and 4 highest. Fisher's exact test p=3.6 x 10$^{-5}$.

The following figure supplements are available for figure 6:

**Figure supplement 1.** Effect of rapamycin on the fecal microbiome.

**Figure supplement 2.** Bacterial DNA in fecal samples significantly different between control and rapamycin.

**Figure supplement 3.** Bacterial composition in fecal samples at phylum level.

## Discussion

Taken together, our data demonstrate that a single three-month regimen of rapamycin is sufficient to robustly increase life expectancy in middle-aged mice, comparable to the effects previously

reported for life-long treatment, while also improving measures of healthspan and substantially altering the microbiome. This work extends prior evidence indicating that short-term rapamycin treatment can improve health in mice, including one experiment suggesting that 4 mg/kg rapamycin every other day for 6 weeks enhanced survival until around 30 months of age in a small cohort (*Chen et al., 2009*), and studies reporting improvements in cardiac (*Dai et al., 2014*; *Flynn et al., 2013*) and immune (*Chen et al., 2009*) function following transient treatment with rapamycin. In the animals treated with the 126 ppm eRapa diet in this study, the improvements in lifespan and health were achieved without overt detrimental side effects, although it is possible that some side effects were undetected, and we did not explicitly test for cataracts, gonadal degeneration, and other adverse outcomes. In the case of the 8 mg/kg/day injection regimen, serious side effects were noted in female, but not male mice. Intriguingly, a dramatic shift toward aggressive hematopoietic cancers and away from non-hematopoietic cancers was observed in these female mice. This is consistent with a similar weak trend seen in kidney transplant patients receiving rapamycin to prevent organ rejection, suggesting a possible conservation of mechanism and clinical relevance (*Mathew et al., 2004*). Our data indicate a need to carefully consider sex effects when optimizing treatment regimens and mechanism of drug action. They also illustrate the importance of better understanding the effects of mTOR inhibitors on differential cancer risk, particularly as mTOR inhibitors are being tested and used clinically for a variety of purposes including the treatment of some rare forms of cancer. The importance of evaluating potential risks and adverse side effects when developing interventions to promote healthy aging should not be underestimated.

This study extends other recent work aimed at developing mid-life interventions to promote healthy aging. Of particular note are two studies reporting improved healthspan from short-term treatments in older mice with a JAK pathway inhibitor (*Xu et al., 2015*) and increased lifespan after transient treatment with the NAD$^+$ precursor nicotinamide riboside (*Zhang et al., 2016*). From a translational perspective, a healthy aging intervention that can be applied for a relatively short period of time during mid- or late-life is likely to have advantages in cost, practicability and quality of life of people, and we look forward to further developments in this area.

## Materials and methods

### Mouse lifespan studies

All lifespan and healthspan experiments were performed on 19–20 month old C57BL6/JNia obtained from the National Institute on Aging Aged Rodent Colony. A separate cohort of C57BL6/J ranging from 17 weeks to 100 weeks of age was obtained from the Harrison lab at the Jackson Laboratory for histological analysis of SFB. Animals were housed in individually ventilated cages (Allentown, Allentown, NJ) containing corncob bedding (Andersons, Maumee, OH) and nestlets. Mice were fed irradiated Picolab Rodent Diet 20 #5053 (Lab Diet, St. Louis, MO). Animals were maintained in a specific pathogen free facility within a *Helicobacter spp.*-free room. Mice were housed in groups (5 per cage at a maximum) and aggressive male mice were isolated to prevent fighting. Mice were acclimatized at least two weeks before onset of experiments. Mice were inspected daily, and medicated for non-life threatening conditions as directed by the veterinary staff. Date of death was recorded when mice were found dead or unlikely to survive longer than 48 hr at the time of inspection. Mice were euthanized according to the following criteria (modified from the Intervention Testing Program protocol [*Harrison et al., 2009*]) when they showed one of these symptoms: (1) inability to eat or drink, (2) severe lethargy, as indicated by a lack of response such as a reluctance to move when gently prodded, (3) severe respiratory difficulty while at rest, indicated by a regular pattern of deep abdominal excursions or gasping, or showing any combination of the following features: (a) severe balance and gait disturbance, (b) an ulcerated or bleeding tumor, visible to the naked eye and breaking through the skin of the animal, or rapid weight gain associated with visible or palpable masses, or (c) Body Condition Score equal to 1 or loss of 20% of body weight in the course of seven days. Survival data is shown from the end of treatment (23–24 months of age). Deaths before and during the three month treatment period by group were as follows: 2 vehicle-injected males, 3 rapamycin-injected males, 1 eudragit fed female, 1 eRapa fed male. Sentinel mice (Crl:CD1[ICR]; Charles River, Wilmington, MA) were tested quarterly and were negative for endo/ectoparasites, mouse norovirus, mouse hepatitis virus, mouse parvovirus, and rotavirus. Sentinel mice were tested annually for

Mycoplasma pulmonis, pneumonia virus of mice, reovirus 3, Sendai virus, and Theiler murine encephalomyelitis virus. All care of experimental animals was in accordance with the University of Washington institutional guidelines and experiments were performed as approved by the Institutional Animal Care and Use Committee.

For the injection experiments, rapamycin (LC Laboratories) was dissolved in DMSO to 100 mg/ml, then further diluted in 5% PEG-400/5% Tween-80 to a final concentration of 1.2 mg/ml, sterile filtered, and stored at −80°C for long-term storage. 37 rapamycin treated mice (17 males, 20 females) were i.p. injected daily for 3 months with 66 µl/10g body weight for a final dosage of 8.0 mg/kg starting at 20–21 months old. 38 Control mice (18 males, 20 females) were i.p. injected with vehicle solution (5% PEG-400/5% Tween-20) for 3 months. For the feeding model, encapsulated rapamycin was obtained from Rapamycin Holdings, Inc. Irradiated PicoLab Diet 20 5053 pellets were ground and mixed with encapsulated rapamycin at 126 ppm. 300 ml of 1% agar melted in sterile water and 200 ml of sterile distilled water were added per kilogram of powdered chow, in order to make pellets. Pellets were stored at −20°C until use. Control food contained the same concentration of agar and encapsulation material (eudragit) without rapamycin at the concentration that matched the rapamycin chow. 37 mice (19 on eudragit, 18 on rapamycin) received assigned diet treatments at 20–21 months of age, lasting for 90 days.

## Body weight and food intake measurements

Body weight was measured at least weekly from the beginning of treatment until three months after the end of treatment. For male mice receiving injections (*Figure 1A*), at each time point from the start point, sample size was as follows: vehicle N = 19, 18, 18, 18, 18, 18, 18, 18, 18, 18, 18, 18, 18, 17, 17, 17, 17, 17, 17, 17, 17, 17, 16, 16, 16, 16, 14, 13, 13, 13, 13, 13, 13; rapamycin N = 20, 19, 19, 19, 19, 19, 19, 19, 19, 19, 18, 18, 18, 17, 17, 17, 17, 17, 17, 17, 17, 17, 17, 17, 16, 16, 16, 16, 16, 16, 16, 15, 15. For female mice receiving injections (*Figure 2A*), at each time point from the start point, sample size was as follows: vehicle N = 20, 20, 20, 20, 20, 20, 20, 20, 20, 20, 20, 20, 20, 20, 20, 20, 19, 19, 19, 19, 18, 18, 18, 18, 17, 17, 17, 16, 15, 15, 13, 12, 11 vehicle; rapamycin N = 20, 20, 20, 20, 20, 20, 20, 20, 20, 20, 20, 20, 20, 20, 20, 20, 19, 18, 18, 18, 18, 17, 17, 17, 16, 16, 16, 16, 15, 14, 13, 13. For mice receiving micro-encapsulated diets (*Figure 4*), at each time from the start point, sample size was as follows: eudragit males N= 20, 20, 20, 20, 20, 20, 20, 20, 20, 20, 20, 20, 20, 20, 20, 20, 20, 19, 18, 18, 18, 18, 18, 18, 18; eudragit females N= 19, 19, 19, 19, 19, 19, 19, 19, 19, 19, 19, 19, 19, 19, 18, 18, 18, 18, 18, 17, 17, 17, 17, 16, 16, 15, 15; eRapa males N= 19, 19, 19, 19, 19, 19, 19, 19, 19, 18, 18, 18, 18, 18, 17, 17, 17, 17, 17, 17, 17, 17, 17, 17, 17, 17, 17; eRapa females N= 18, 18, 18, 18, 18, 18, 18, 18, 18, 18, 18, 18, 18, 17, 17, 17, 17, 17, 17, 16, 16, 16, 16, 15, 15, 15.

Food intake was measured weekly or twice per week per cage by subtracting the amount of food remaining on the wire rack from the amount given 4–7 days before. Average food intake per mouse was calculated by dividing this value by the number of mice in the cage, then averaging all the values from cages hosting mice under the same treatment and of the same sex.

## Histopathological analysis

Gross examination was performed following the natural death or euthanasia of animals. Tissues from mice were fixed with 10% neutral buffered formalin, routinely processed and embedded in paraffin, and stained with haematoxylin and eosin (H&E). 16 rapamycin injected female, 15 rapamycin injected male, 12 vehicle administered female, and 14 vehicle administered male counterparts in the lifespan study were analyzed for tumor incidence and extent by two board certified veterinary pathologists (J.M.S. and P.M.T). 7/15 rapamycin injected male mice were autolyzed to varying degrees, which may have complicated histological detection of systemic neoplasia, although disease sufficient to result in death was determined for 3 of these mice and systemic neoplasia was detected on histological examination of 5 of these 7 mice. Autolysis also complicated the histological assessment of 2 female vehicle treated mice, although systemic neoplasia was detected on histological examination of one of these two mice. Major organs (including decalcified cross section of the head, skin, lung, heart, liver, kidney, spleen, pancreas, lymph node, salivary gland, gastrointestinal tract and reproductive tract) were examined histologically.

For semi-quantitative SFB analysis, 14 female and 16 male mice i.p. injected with rapamycin (8 mg/kg/day, daily for 3 months) and 9 female and 17 male mice injected with vehicle (daily for 3

months) were sacrificed the day after the end of treatment, and H&E sections of the gastrointestinal tract were routinely prepared and examined. The amount of SFB in the intestinal tissue was graded on a 0 to 4 scale, with 0 representing no SFB, 1 representing rare SFB, 2 representing mild colonization by SFB, 3 representing moderate colonization by SFB, and 4 representing severe colonization of the small intestine by SFB.

Images of representative lesions were acquired using NIS-Elements BR 3.2 64-bit and plated in Adobe Photoshop Elements. Image brightness and contrast was adjusted using Auto Smart Fix and Auto White Balance manipulations applied to the entire image. Original magnification is stated.

## Rapamycin serum level measurements

Fresh blood was collected upon euthanasia via cervical dislocation 24 hr after the last injection, and sera were isolated using serum separators tube (BD, Franklin Lakes, NJ) and immediately stored at $-20$ C. Rapamycin was extracted with 100 mM $ZnSO_4$ at room temperature and analyzed using a Waters 2795 LC/QuattroMicro MS (Waters, Milford MA).

## Rotarod assay

Mice were tested on a Rotamex V rotarod (Columbus Instruments, Columbus OH) with a constant acceleration of 0.1 rpm/second over a period of four days. On the first day, all animals were allowed to acclimate to the rotarod with a single round of testing. Over the following three days (labeled day 1, 2, and 3) all mice were tested three times, with a 30 min minimum resting period in between rounds. All mice were subjected to rotarod testing prior to initiation of treatment, upon cessation of treatment, and 90 days after cessation of treatment. Individuals running the rotarod test were not blinded to the treatment.

## Forelimb grip strength test

Forelimb grip strength was tested with a Chatillon DFE-050 force gauge (AMETEK, Largo FL). Animals were held by the base of the tail, allowed to grip the bar of the gauge, and slowly pulled away from the testing apparatus with a smooth horizontal movement. Each mouse was tested 5 times per round, and maximum grip strength was recorded. Animals were tested prior to initiation of treatment, upon cessation of treatment, and 90 days after cessation of treatment. At least 72 hr rest was allowed between grip strength and rotarod testing. Individuals running the grip strength test were not blinded to the treatment.

## Microbiome analysis

All fecal samples were collected per cage at 3 months of treatment immediately after excretion from the mice analyzed for lifespan analysis and frozen with liquid nitrogen. Microbial DNA was extracted as previously described (*Ericsson et al., 2015*). Briefly, fecal samples were collected into 800 μL of lysis buffer (500 mM NaCl, 50 mM tris-HCl, 50 mM EDTA, and 4% SDS), homogenized in a Qiagen Tissuelyser II (Valencia CA), and incubated at 70°C for 20 min. The supernatant was mixed with 1 μL of 2 M ammonium acetate, incubated on ice, and then centrifuged at 16,000 × g for 10 min at room temperature. The supernatant was then mixed with an equal volume of ice-cold isopropanol and incubated for 30 min on ice. The contents of the tube were then centrifuged at 4°C for 15 min to pellet DNA. The pellet was rinsed twice with 70% EtOH and re-suspended in 150 μL of tris-EDTA buffer. DNA was further purified using DNeasy kit (Qiagen, Valencia CA) according to the manufacturer's protocol. For Metagenomic sequencing, sequencing of the V4 region of the 16S rRNA gene was performed on the Illumina MiSeq platform (San Diego CA), as previously described (*Ericsson et al., 2015*). The raw metagenome data are publicly available at the European Nucleotide Archive (ENA) database (ERP014805).

## Real-time PCR

Real-time PCR to measure SFB and total bacterial DNA was performed on a CFX384 Real-Time System with a C1000-Touch thermal cycler (BioRad, Hercules, CA) and a StepOnePlus (Applied Biosystems, Foster City, CA) with a Sybr Green method as previously described (*Ericsson et al., 2015*). 10 ng of DNA collected from fecal samples was analyzed. Primers for SFB are 5' TGTGGGTTGTGAA TAACAAT 3' and 5' GCGAGCTTCCCTCATTACAAGG 3'. Primers for the detection of total bacteria

(16s rRNA gene) are 5' TCCTACGGGAGGCAGCAGT 3' and 5' GGACTACCAGGGTATCTAATCCTG TT 3'.

## Statistical analysis

An unpaired two-sample t test was used to compare two experimental groups unless otherwise mentioned. Post-treatment survival data were analyzed using a one-sided Mann Whitney U test (http:// vassarstats.net/utest.html). Fisher's exact test was used for semi-quantitative analysis of SFB and the incidence of malignancies. Rotarod p-values were calculated by applying the *glht()* function for general linear hypotheses for mixed-effects models from the R multcomp package (*Hothorn et al., 2008*) to each of the outputs from the *lmer()* function in the R lme4 package (*Bates et al., 2014*). For the differential abundance analysis of microbiome population, we used the fitZig() function in the R package metagenomeSeq (*Paulson et al., 2013*).

The *Figure 6—figure supplement 1* heatmap of hierarchical clustering of 16S rRNA gene prevalence was generated using Euclidian Distance using the R package pheatmap. The shift in the microbial population after controlling for the experimental effect was tested using distance matrix based permutation MANOVA implemented in R package vegan as the adonis() function using the weighted Unifrac distance. Violin plots and stacked barplot were created using R package ggplot2. In the above analysis we removed a library that exhibited signs of experimental failure (total library size <10000).

## Acknowledgements

We thank members of the Kaeberlein lab for technical assistance and helpful discussion. Especially, we thank Jeehae Han, Hillary Miller, Fresnida Ramos, Melana Yanos, Chenhao Lu, Jacob Kahn, Quy Nguyen and Oliver Tamis for help with animal experiments. We thank Kenneth Chen for critical reading of the manuscript. We thank the Jackson Laboratories Aging Center and Nathan Shock Center of Excellence for providing C57BL/6J mice used for the histopathology studies. We thank David Harrison, the Jackson Laboratories, Terri Iwata and Thea Brabb for their assistance in drafting the endpoint criteria for the lifespan studies. We thank Terri Iwata and Ryan Centini for helping with necropsies for lifespan cohort animals. We thank the Foege Veterinary Services and animal husbandry staff for their assistance in the care of the animals. We thank Aika Nojima for helping to inspire the project by sharing her unpublished data on transient rapamycin injection models. This work was supported by a grant to MK from the Samsung Electronics Company, by the University of Washington Nathan Shock Center of Excellence in the Basic Biology of Aging (NIH P30AG013280) and the University of Washington Healthy Aging and Longevity Research Institute. AB was supported by NIH training grant T32AG000057. TKI was supported by JSPS Postdoctoral Fellowship and Uehara Memorial Foundation Postdoctoral Research Fellowship.

## Additional information

### Funding

| Funder | Grant reference number | Author |
|---|---|---|
| National Institute on Aging | T32AG000057 | Alessandro Bitto |
| Japan Society for the Promotion of Science | | Takashi K Ito |
| Uehara Memorial Foundation | | Takashi K Ito |
| Samsung | | Matt Kaeberlein |
| National Institute on Aging | P30AG013280 | Matt Kaeberlein |
| University of Washington | | Matt Kaeberlein |

The funders had no role in study design, data collection and interpretation, or the decision to submit the work for publication.

## Author contributions
AB, Designed experiments, Wrote the paper, Performed drug injections for mice for semi-quantitative grading of SFB, Performed lifespan experiments on feeding models, Performed rotarod assay and forelimb grip strength test; TKI, Conceived the project, Designed experiments, Wrote the paper, Performed lifespan experiments on injection models, Performed analysis on fecal sample size, Performed microbiome analysis, Performed the quantitative analysis of SFB with real-time PCR; VVP, Performed the quantitative analysis of SFB with real-time PCR, Drafting or revising the article; NJL, Performed analysis on fecal sample size, Contributed unpublished essential data or reagents; HZH, Performed lifespan experiments on injection models, Drafting or revising the article, Contributed unpublished essential data or reagents; ES, HT, NV, Performed drug injections for mice for semi-quantitative grading of SFB, Performed lifespan experiments on feeding models, Drafting or revising the article; BC, KS, DM, Performed lifespan experiments on feeding models, Drafting or revising the article; MY, Performed microbiome analysis, Performed statistical analysis and designed the figures on microbiome analysis, Drafting or revising the article, Contributed unpublished essential data or reagents; RPB, KFK, Performed statistical analysis and designed the figure on rotarod assay and part of lifespan analysis, Drafting or revising the article, Contributed unpublished essential data or reagents; DJD, Performed microbiome analysis, Performed the quantitative analysis of SFB with real-time PCR, Drafting or revising the article; CHG, Performed microbiome analysis, Provided intellectual input in SFB analysis, Drafting or revising the article; JMS, Wrote the paper, Performed histopathological analysis and semi-quantitative grading of SFB; PMT, Designed experiments, Performed histopathological analysis and semi-quantitative grading of SFB, Provided intellectual input in SFB analysis, Drafting or revising the article; MK, Conceived the project, Designed experiments, Wrote the paper, Managed and supervised the whole project

## Author ORCIDs
Matt Kaeberlein, http://orcid.org/0000-0002-1311-3421

## Ethics
Animal experimentation: This study was performed in accordance with the recommendations in the Guide for the Care and Use of Laboratory Animals of the National Institutes of Health. All of the animals were handled according to approved institutional animal care and use committee (IACUC) protocols (#4359-01) of the University of Washington.

## Additional files

### Major datasets
The following dataset was generated:

| Author(s) | Year | Dataset title | Dataset URL | Database, license, and accessibility information |
| --- | --- | --- | --- | --- |
| Ito TK, Yajima M | 2015 | Transient rapamycin treatment robustly increases lifespan and healthspan in middle-aged mice | http://www.ebi.ac.uk/ena/data/view/ERP014805 | Publicly available at the EBI European Nucleotide Archive (accession no: ERP014805) |

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
