## [Decision Letter]

Thank you for submitting your article "Transient rapamycin treatment increases lifespan and healthspan in middle-aged mice" for consideration by *eLife*. Your article has been reviewed by three peer reviewers, one of whom, Amy Wagers, is a member of our Board of Reviewing Editors, and the evaluation has been overseen by Kevin Struhl as the Senior Editor.

The reviewers have discussed the reviews with one another and the Reviewing Editor has drafted this decision to help you prepare a revised submission.

Summary:

The manuscript by Kaeberlein and colleagues investigates the impact on lifespan of transient exposure of older mice to rapamycin (via injection or as a dietary supplement). Assessment of some healthspan metrics (grip strength and rotorod performance) is also included. The authors present intriguing data showing sex-specific differences with high dose rapamycin in terms of lifespan extension and effect on cancer development, and congruent effects on longevity in both sexes with lower dose, oral administration. Finally, they report differences in the composition of gut microbiota that are associated with rapamycin administration. Reviewers agreed that the manuscript has a number of strengths, but also several weaknesses.

In particular, reviewers appreciated the careful assessment of lifespan and cause of death in both sexes, the relatively novel approach of transient rapamycin administration in pre-senescent mice, and the evaluation of rapamycin at different doses and via different routes of administration, which reveal novel information regarding the potential benefits and possible toxicities of rapamycin administration for the treatment of age-associated dysfunctions. Furthermore, the association with changes in microbiota composition present some intriguing potential explanations for rapamycin's effects. Yet, this aspect of the study also represents its major limitation, as the authors do not provide any direct link between SFB colonization and lifespan extension. This issue is exacerbated by the fact that the aged mice used for analysis of SFB colonization and lifespan studies derive from different animal colonies. The inclusion of additional data that strengthen the connection between SFB, rapamycin and lifespan regulation is needed. In addition, it is essential to assess plasma levels of rapamycin achieved by the different dosing regimens in male vs. female mice.

Essential revisions:

1) The major problem in this study is the lack of compelling support for the importance of microbiota for the rapamycin-induced healthspan extension. The association of short-term Rapa treatment with increased SFB is interesting, but the authors have not provided a causal link between the increase in this organism and any of the phenotypes observed. A direct link should be demonstrated experimentally through addition of new experiments, e.g. fecal transfer from SFB monoassociated mice and/or intervention to specifically deplete SFB (if possible).

2) An additional concern is that the aged mice used for analysis of SFB colonization and lifespan studies derive from different animal colonies (see Methods), making the connection between these observations more tenuous. Confirmation of SFB differences in NIA mice should be added to the manuscript to validate the inferences made in the text.

3) For Figure 6, it is unclear if the rapa treatment group represents injected or fed rapamycin, and if males and females are separately analyzed. Please clarify the text on this point. Also, as dietary intake of eRAPA could have distinct effects on microbiota compared to ip injection, the authors should add to the manuscript a comparison of microbiota analyses between these two administration methods.

4) Since plasma rapamycin levels following oral administration differed between males and females in the Intervention Testing Program, it is important to know the plasma levels achieved following the oral and ip doses given in the current study. This could be part of the reason for the sex effect observed. The authors should add analysis of plasma levels achieved by oral or ip dosing in males and females in the study. Also, as differences in food intake or physical activity/energy expenditure could also provide a possible explanation, measurements of these variables should be added to the manuscript for the two different administration methods.

5) What were major causes of death in males ip-injected with rapamycin and in both males and females administered with eRAPA orally? Were there any particular causes of death in each cohort? The authors should add this information to the manuscript.

6) Changes in rotarod performance over time are not adequate to support the improvement in memory formation by transient rapamycin treatment. The authors should tone down their description or conduct appropriate experiments.

7) Some figures are inaccurately referenced in the text. The authors should check these carefully to ensure accuracy.

[Editors' note: further revisions were requested prior to acceptance, as described below.]

Thank you for resubmitting your work entitled "Transient rapamycin treatment increases lifespan and healthspan in middle-aged mice" for further consideration at *eLife*. Your revised article has been favorably evaluated by Kevin Struhl (Senior editor) and a Reviewing editor.

The manuscript has been improved, and most issues have been addressed by new data, revision/clarification of the text, or by explaining why these issues cannot be addressed at the present time. However, there are some remaining issues that need to be addressed before acceptance, as outlined below:

1) Please edit the title. Since lifespan increases are shown to be sex- and dose-variant in this study, the title should be changed to "Transient rapamycin treatment can increase lifespan and healthspan in middle-aged mice". This will help to emphasize the very important points the authors raise in their discussion about evaluating potential risks and side effects of drug interventions for lifespan regulation.

2) In the new data added in Figure 1—figure supplement 2 or Figure 2—figure supplement 2, why are there no error bars?

---

## [Author Response]

*Essential revisions:*

*1) The major problem in this study is the lack of compelling support for the importance of microbiota for the rapamycin-induced healthspan extension. The association of short-term Rapa treatment with increased SFB is interesting, but the authors have not provided a causal link between the increase in this organism and any of the phenotypes observed. A direct link should be demonstrated experimentally through addition of new experiments, e.g. fecal transfer from SFB monoassociated mice and/or intervention to specifically deplete SFB (if possible).*

We appreciate this concern, and completely agree that addressing the causality of microbiome changes in longevity is of primary interest moving forward. We have attempted to state clearly in the text that we are not claiming causality in this manuscript and, we respectfully note that such an experiment would take more than a year to complete and perhaps up to two years to show effects on survival. However, it is not even currently possible for us to do this, as we are no longer able to get aged C57BL/6Nia mice from the NIA aged rodent colony because they have changed their rules such that an investigator must have a NIA grant approved for vertebrate animals in order to get access to these animals. Despite attempts to get such support from NIA, we do not currently have an NIA grant that qualifies for access to these animals.

*2) An additional concern is that the aged mice used for analysis of SFB colonization and lifespan studies derive from different animal colonies (see Methods), making the connection between these observations more tenuous. Confirmation of SFB differences in NIA mice should be added to the manuscript to validate the inferences made in the text.*

We apologize for not being clearer in our presentation. Increased SFB colonization was observed independently both in the cohort obtained from the Jackson Laboratories (histologically) and in the cohorts obtained from NIA and used for lifespan analysis (via sequencing and realtime PCR). In addition, this increase in SFB associated with rapamycin treatment was observed both in the dietary delivery regimen (NIA mice) and in the daily injection regimen (both JAX and NIA mice). Thus, the effect was reproduced with animals obtained from mouse colonies from two different locations using two different rapamycin delivery methods. We have amended the text of both the results and methods section to make this clearer.

3) For Figure 6, it is unclear if the rapa treatment group represents injected or fed rapamycin, and if males and females are separately analyzed. Please clarify the text on this point. Also, as dietary intake of eRAPA could have distinct effects on microbiota compared to ip injection, the authors should add to the manuscript a comparison of microbiota analyses between these two administration methods.

The rapamycin treatment group includes all cohorts for the injection and feeding model. Fecal samples were collected per cage from all the lifespan cohort cages. We have clarified this in the text. The effects of delivery method, sex, batch differences (mice purchased at a different time), and rapamycin are analyzed in Figure 6—figure supplement 1, supplement 2B, and 2C. As shown in Figure 6—figure supplement 1, the delivery method (injections with solvents or food containing capsules and agar) showed the largest effect on the composition of the gut bacteria. Considering multiple factors, SFB and other strains were identified as significantly more abundant in all rapamycin treated samples as shown in Figure 6 and Figure 6—figure supplement 2.

*4) Since plasma rapamycin levels following oral administration differed between males and females in the Intervention Testing Program, it is important to know the plasma levels achieved following the oral and ip doses given in the current study. This could be part of the reason for the sex effect observed. The authors should add analysis of plasma levels achieved by oral or ip dosing in males and females in the study. Also, as differences in food intake or physical activity/energy expenditure could also provide a possible explanation, measurements of these variables should be added to the manuscript for the two different administration methods.*

We are unable to provide blood levels of rapamycin from the mice used for lifespan analysis. However, no statistically significant differences were noted in the trough levels (24 hours post injection) in the separate cohort injected with 8 mg/kg/day i.p. for 3 months and sacrificed for histological analysis. The following chart, reporting these measurements is inserted as Figure 1—figure supplement 1 as well as included here for the reviewer. Charts describing food intake for the mice on both delivery methods have been inserted as supplements to Figure 1, Figure 2, and 4.

*5) What were major causes of death in males ip-injected with rapamycin and in both males and females administered with eRAPA orally? Were there any particular causes of death in each cohort? The authors should add this information to the manuscript.*

We have performed end of life pathology on the male mice injected with rapamycin and their vehicle injected counterparts. Results of the analysis have been added to the manuscript and are reported below for the reviewer’s convenience. Regretfully, we are unable to perform a thorough analysis of the mice fed eRapa or eudragit, although gross examination at time of necropsy did not reveal any clear difference between control and treated animals or between males and females. We hope to be able to perform for comprehensive analyses of disease burden at time of death for different doses of eRAPA in the future, pending availability of funding.

The following text has been added to the manuscript: “Both control and rapamycin treated male cohorts had a high incidence of systemic round cell neoplasia and most of them affected ≥2 organs, and had an intravascular component and a morphology most consistent with histiocytic sarcoma, although immunohistochemistry was not performed (Figure 3—figure supplement 2). […] In the male group, no cases of systemic lymphoma with plasmacytoid morphology were seen.”

*6) Changes in rotarod performance over time are not adequate to support the improvement in memory formation by transient rapamycin treatment. The authors should tone down their description or conduct appropriate experiments.*

We have removed reference to memory formation and the sentence “consistent with prior studies reporting the improvement of cognitive function in mice treated with rapamycin (*4, 5*)” has been removed from the text.

7) Some figures are inaccurately referenced in the text. The authors should check these carefully to ensure accuracy.

We apologize for this. We have carefully reviewed the text and made the necessary corrections.

[Editors' note: further revisions were requested prior to acceptance, as described below.]

*1) Please edit the title. Since lifespan increases are shown to be sex- and dose-variant in this study, the title should be changed to "Transient rapamycin treatment can increase lifespan and healthspan in middle-aged mice". This will help to emphasize the very important points the authors raise in their discussion about evaluating potential risks and side effects of drug interventions for lifespan regulation.*

We agree with this comment and we have changed the title of the manuscript accordingly.

*2)In the new data added in Figure 1—figure supplement 2 or Figure —figure supplement 2, why are there no error bars?*

These graphs report average food consumption per mouse, but food consumption was measured per cage at each time point, as described in the method section “body weight and food intake measurements”. We obtained average food consumption values per mouse by dividing the measured intake per cage by the number of mice in each cage, then averaging these values across cages receiving the same treatment. Because of these calculations, we believe that error bars representing standard deviation or standard error would be meaningless and not add any valuable information to these plots.